# Tongue and Upper Airway Dimensions: A Comparative Study between Three Popular Brachycephalic Breeds

**DOI:** 10.3390/ani11030662

**Published:** 2021-03-02

**Authors:** Johannes Sebastian Siedenburg, Gilles Dupré

**Affiliations:** University Clinic for Small Animals, Small Animal Surgery, Department for Companion Animals and Horses, University of Veterinary Medicine, 1210 Vienna, Austria; gillespierre.dupre@gmail.com

**Keywords:** macroglossia, brachycephalic obstructive airway syndrome, pug, French bulldog, English bulldog, computed tomography

## Abstract

**Simple Summary:**

Brachycephalic obstructive airway syndrome is a debilitating disease complex, that affects severely brachycephalic dogs, impairs their quality of life and shortens life expectancy. Macroglossia has been identified as part of the soft tissue conditions that characterize brachycephalic breeds. Despite similar degrees of brachycephalism comparing the common breeds French bulldogs, English bulldogs and pugs, several breed specific characteristics contributing to brachycephalic obstructive airway syndrome have been described in the past. The present study aimed to examine the tongue volume and at three levels cross-sectional areas of the tongue, oropharyngeal airways, soft palate and nasopharyngeal airways in the aforementioned breeds. Assessment of computed tomography scans revealed smaller tongues in pugs compared to French and English bulldogs, with values being close to previously published data in mesaticephalic breeds. Comparing the cross-sectional areas between breeds, several differences were identified, however, calculating the impact of soft tissue on total airways areas uncovered only differences between pugs and French bulldogs at the most caudal location, where the latter breed had a greater ratio, presumably in consequence to a significantly larger oropharyngeal airway area. These findings corroborate the importance of respecting breed specific differences in regard to surgical treatment of brachycephalic obstructive airway syndrome.

**Abstract:**

Relative macroglossia has been identified in brachycephalic compared to mesaticephalic dogs. This study assessed the tongue volume comparing three common brachycephalic breeds, including 30 French bulldogs, 30 pugs, and 15 English bulldogs. Computed tomography scans of intubated dogs with the hard palate suspended were examined for total tongue volume and cross-sectional areas (CSAs) of the tongue, oropharynx, soft palate, and nasopharyngeal airways at three levels: 1, caudal tip of the hard palate; 2, caudal tip of the hamulus; 3 cranial to the basihyoid bone. Tongue volume normalized to bodyweight, was significantly higher in English and French bulldogs than in pugs. Normalized to skull length, CSA of the tongue was smaller in pugs than in French and English bulldogs. At level 3, French bulldogs had larger oropharyngeal CSA than English bulldogs and pugs. Soft palate CSA was the largest in English bulldogs at level 3. At levels 1 and 2, soft palate and nasopharyngeal CSA was the smallest in pugs. At level 3, French bulldogs had higher total airway/soft tissue ratios than pugs. The smaller tongue volume in pugs questions the accuracy of the term macroglossia in this breed and these findings should be considered if surgical correction is sought.

## 1. Introduction

Brachycephalic obstructive airway syndrome (BOAS) is commonly encountered in several breeds, especially in extreme brachycephalic dogs such as French bulldogs, pugs, and English bulldogs, due to a variety of congenital anatomical properties differing from mesaticephalic breeds [1,2,3,4,5,6,7]. Compared with mesaticephalic and dolichocephalic breeds, the skull conformation in brachycephalic dogs is wider and shorter, showing significant, complex, breed-specific shortening of the craniofacial bones accompanied by pronounced alterations of the nasal cavities and pharyngeal soft tissue conformation [1,6,8,9,10]. BOAS is associated with the stenotic nares; aberrant nasopharyngeal turbinates; increased prevalence of nasal mucosal contact points, a thickened and elongated soft palate, which contributes to narrowing of the nasopharyngeal airways; and nasopharyngeal mucosal hypertrophy in addition to tonsillar eversion and hypertrophy [5,11,12,13,14,15,16]. Concurrent tracheal hypoplasia is of embryologic origin rather than a consequence of rostral airway obstruction [10]. 

It is deduced that forced inspiration and turbulent air flow lead to increased negative luminal pressure in the pharynx and larynx, in consequence of the chronic upper airway obstruction, promoting mucosal edema, swelling, eversion of laryngeal saccules, and laryngeal collapse [4,10,17]. Bronchial collapse, less commonly described as a consequence of BOAS, was significantly associated with the degree of laryngeal collapse; pugs were most commonly and severely affected [18]. In addition, a high incidence of gastrointestinal anomalies and clinical signs such as regurgitation and vomiting have been described in dogs affected by BOAS [19,20,21]. 

Clinical signs resulting from the aforementioned conditions delineating BOAS range from snoring, significantly increased inspiratory effort, chronic hypoxemia, syncopal episodes, and cyanosis to more sleep-disordered breathing-related signs such as irregular breathing patterns, apnea, and oxygen desaturation [22,23,24,25]. 

Previously published articles have shown significant differences between the most common brachycephalic breeds regarding clinical characteristics of BOAS. For instance, French bulldogs often present with gastrointestinal signs, which are less commonly encountered in pugs [13]. Anatomical and functional differences are either linked to BOAS, pugs have smaller relative nasopharyngeal CSAs, or characteristics differentiating brachycephalic breeds such as minuscule frontal sinuses, and rostrally rotated maxilla in pugs [21,26]. While English bulldogs are most commonly affected by a hypoplastic trachea, pugs were more frequently affected by severe laryngeal collapse than French bulldogs [27,28]. 

The overly long and thick tongue in brachycephalic breeds has been repeatedly mentioned, contributing to a dorsal displacement of the soft palate, causing narrowing of the nasopharynx [10,14,29]. However, until recently, there has been a paucity of scientific descriptions concerning the contribution of a relatively large tongue (macroglossia) to the oropharyngeal soft tissues in brachycephalic dogs [30]. This study demonstrated an increased normalized total tongue volume in brachycephalic when compared to mesaticephalic breeds and accordingly outlined a decrease in total air area indexed to the total soft tissue area at certain levels in the oropharynx and nasopharynx in brachycephalic dogs [30]. To our knowledge, the degree of macroglossia in different brachycephalic breeds has not been characterized. 

For normalizing morphometric data accounting for different sizes between breeds, several indices have been proposed, such as the LW index (skull length/skull width), bodyweight, and skull length [2,9,30,31]. However, to the best of the authors knowledge, no consensus exists regarding appropriate normalization methods neither for tissue volumes, nor cross sectional areas of parts of the skull in small animals. The aim of the present study was to objectively compare the total tongue volume of three common brachycephalic breeds and to evaluate the relationship between total soft tissue and total airway dimensions in the oropharynx and nasopharynx. It was hypothesized that total tongue volume normalized to bodyweight would differ between the three brachycephalic breeds. Furthermore, pugs would have smaller nasopharyngeal cross-sectional areas (CSAs), leading to no differences in total air to total soft tissue ratios between the three breeds.

## 2. Materials and Methods

### 2.1. Materials

Client-owned French bulldogs, pugs (30 of each breed) and English Bulldogs (15 dogs), which presented for clinical signs of BOAS, were retrospectively enrolled in the study. Inclusion criteria comprised a full clinical examination, a CT examination of the head, an endoscopic examination of the upper airways, and a corrective multilevel upper airway surgery. Cases who underwent any type of BOAS-related surgery previously and those with any non-BOAS-related pathologies such as neoplastic or inflammatory masses in the upper airways or surrounding soft tissues were excluded from the study.

### 2.2. Methods

History, clinical signs, sex, age, breed affiliation, and weight of the dogs were obtained at admission. All dogs underwent a complete general physical examination and basic blood investigations (packed cell volume [PCV]/total solids [TS] if ≤6 months old and complete blood count and serum biochemistry if >6 months of age) as work-up before anesthesia for CT scans of the head and neck. Food and water were withheld for at least 12 h before admission to the hospital. Anesthetic regimen was adapted to requirements of individual cases with varying combinations of the following drugs: methadone (0.2 mg/kg intravenous [IV]), midazolam (0.2–0.3 mg/kg IV), acepromazine (10–15 µg/kg IV), ketamine (0.2–0.5 mg/kg intramuscular [IM]), butorphanole (0.2 mg/kg IM), and medetomidine (5–10 µg/kg IM). Induction was achieved using propofol (2–6 mg/kg IV) and after intubation maintenance was performed with inhalant anesthetics (isoflurane or sevoflurane).

#### 2.2.1. Computed Tomography Examination

All dogs were kept intubated (median endotracheal tube size: pugs 5.5, French bulldogs 6.0, English bulldogs 6.5) and placed in sternal recumbency with the skull in a standardized, extended neutral position while the hard palate was elevated using a perforated open acrylic glass cage with a tape or acrylic bar, caudal to the incisors or canine teeth, suspending the maxilla, and a vacuum cushion under the mandible and neck. All CT images were acquired using a 16-slice helical CT scanner (Somatom Emotion; Siemens AG, Erlangen, Germany), with a gantry rotation time of 0.6 s, tube voltage of 120 kV, and tube current ranging from 100 to 130 and 150 mA (for dogs weighing ≤10 kg, 10–30 kg and >30–50 kg, respectively) The field of view was set to enclose the entire skull and neck. After confirmation of proper orientation on scout images, the scan started rostral to the nose, including the whole skull, and was extended to a level caudal to the 5th cervical vertebra. Multiplanar reformatting was performed with a slice thickness of 0.3–1.0 mm (mean values: Pugs 0.493 mm, French bulldogs 0.503 mm, and English bulldogs 0.580 mm) and a slice reconstruction interval of 1–2 mm. The opening angle (°) between the rostral border of the incisor bone and the mandibula, measured at the alveolar margin, with the vertex placed on the transverse plane at the caudal border of the temporomandibular joint were assessed on mid-sagittal images in all dogs. The resulting mean values were 24.47 in English bulldogs, 25.03 in French bulldogs and 29.57 in pugs. 

#### 2.2.2. Tongue Volume and Cross-Sectional Area Assessment

Measurements were made in each dog by the same investigator (JS) with a commercially available scientific visualization and segmenting software for medical purposes (Thermo Scientific Amira Software; Thermo Fischer Scientific, Waltham, MA, USA). To compare the degree of brachycephalism, the skull index (LW; skull length/skull width) was calculated [31]. The LW index was calculated by dividing the length of the skull (sl) measured from the most rostral aspect of the incisive bone by the caudal border of the occipital bone on a mid-sagittal image by the width of the skull (sw) measured at the level of the greatest distance between the outer border of the zygomatic arches on transverse images. 

Assessments of the total tongue volume and cross-sectional areas (CSA) of the tongue, soft palate, nasopharynx, and oropharynx at three levels caudal to the hard palate, were performed on reconstructed transverse and sagittal images as previously reported in extubated brachycephalic and mesaticephalic dogs with minor modifications [30].

Briefly, total tongue volume was assessed within the following anatomical borders: The surface of the tongue was demarcated by the oral cavity craniodorsally and caudodorsally by the air-filled oropharynx or soft palate. The ventral border was represented by the mylohyoid, geniohyoid, and hyoid muscles. Rostrally the tongue margin; caudally, the attachment of the tongue at the basihyoid bone; and laterally, the tongue margin, the mandibular teeth, digastricus muscle, and hyoid apparatus delineated the tongue.

Cross-sectional areas of soft tissues and airways were obtained on transverse planes at three specific sections: (1) at the most caudal CT slice displaying the caudal end of the hard palate, (2) at the most caudal CT slice showing the caudal tip of the hamulae of the pterygoid bone; and (3) exactly five CT slices rostral to the rostral tip of the basihyoid bone (Figure 1). General thresholds for air and soft tissue discrimination were set to −1100 to −100 and −200 to 150 HU, respectively. Outlines of the areas of interest were determined by segmentation techniques, using manual adjustments of the window level when drawing the outlines along the borders of anatomic structures on transverse slices from cranial to caudal on every third CT slice. The interpositioned slices were enclosed by computational interpolation and then manually controlled. In the absence of air in the oropharynx, palatal tissues were laterally defined by straight lines connecting the most ventrolateral aspect of the palatine bone and the dorsolateral margin of the oropharynx or the tongue, respectively. The dorsal margin of the palatal tissues was represented by the palatine bone and maxilla, whereas ventrally, the tongue or air-filled oropharynx represented the border. The soft palate was laterally delineated by two straight lines starting from the most dorsolateral aspect of the hamulus of the pterygoid bone, which concluded on the dorsolateral oropharynx or the dorsolateral margin of the tongue, respectively. The soft palate was dorsally margined by the nasopharynx, ventrally by the oropharynx, and laterally by the medial pterygoid muscle (Figure 2).

Borders of the palatal soft tissues were defined by manually trimming using several density windows to detect subtle changes in the case of contact points to soft tissues such as the tongue.

The CSA of the tracheal tube was measured in the same transverse plane (1, 2, 3) and subtracted from the oropharyngeal CSA.

### 2.3. Statistical Analysis

Based on segmentation data, native and normalized variables were compared between French bulldogs, English Bulldogs, and pugs. The following native data were assessed: age (years), bodyweight (kg), and total tongue volume (mm^3^). To account for the inter-individual sizes, total tongue volume was divided by bodyweight before comparisons. The CSA (mm^2^) of the oropharyngeal and nasopharyngeal airways, soft palate, and tongue were normalized to the product of each dog’s skull length × skull width and total cross-sectional area (CSA nasopharynx + soft palate + oropharynx + tongue at levels 1, 2 and 3 respectively) before statistical comparison between breeds.

To express the relative relation of airways to soft tissue structures, cross-sectional areas at levels 1, 2, and 3 were examined as oropharynx + nasopharynx (total airways) indexed to the tongue and indexed to the tongue + soft palate (total soft tissues), respectively. According to the results of the normality test (Shapiro-Wilk test) and homogeneity of variances, the statistical analysis was performed either with a one-way ANOVA with either Tuckey-Kramer or Dunnett T3 post hoc analysis or Kruskal-Wallis test and Dunn-Bonferroni post hoc analysis. Data are presented as median and range. The level of significance was set at *p* < 0.05. All analyses were performed with statistical and graphic software products (Prism 8 GraphPad Software, San Diego, CA, USA, SPSS (Version 24), IBM, Armonk, Armonk, NY, USA).

### 2.4. Dogs

A total of 75 dogs were evaluated in the study with 30 each of pugs and French bulldogs and 15 English bulldogs, including 17 male (5 neutered) and 13 female (4 neutered) French bulldogs, 15 male (3 neutered) and 15 female (8 neutered) pugs, and 11 male (1 neutered) and 4 female (3 neutered) English bulldogs. The median age was 3.08 years, 5.0 years, and 2.62 years for French bulldogs, pugs, and English bulldogs, respectively. The median bodyweight was 12.40 kg, 8.50 kg, and 22.60 kg and the median LW index was 0.941, 0.980, and 0.973 for French bulldogs, pugs, and English bulldogs, respectively. Skull length and bodyweight were significantly different between the breeds, with English bulldogs having the longest skulls and the highest bodyweight followed by French bulldogs and pugs, whereas LW ratio did not differ significantly between any of the brachycephalic breeds (Table 1).

## 3. Results

### 3.1. Tongue Volume

Total tongue volume normalized to bodyweight, did not differ comparing French and English bulldogs; however, values in both breeds significantly exceeded those of pugs (Table 2). 

### 3.2. Cross-Sectional Areas of Tongue, Oropharynx, Soft Palate and Nasopharynx

CSA of the tongue normalized to skull length × skull width (sl × sw) was significantly larger in French compared to English bulldogs at level 2 (Table 3). 

The CSA of the oropharynx did not differ at levels 1 and 2 between the three breeds, whereas at level 3, French bulldogs had significantly larger oropharyngeal areas than the two other breeds (Figure 3). CSA of the soft palate was smaller in pugs than in French bulldogs and English bulldogs at level 1, whereas, at level 3 pugs and French bulldogs had smaller dimensions than English bulldogs. Pugs had significantly smaller nasopharyngeal CSA than the two bulldog breeds at levels 1. No differences were detected at levels 2 and 3.

Similarly, when normalized to total CSA, Pugs and English bulldogs had a smaller CSA oropharynx at level 3 than French bulldogs. At level 1, pugs had smaller soft palate/total CSA ratios than English bulldogs, whereas at level 3, French bulldogs and English bulldogs differed significantly. Pugs demonstrated smaller CSA of the nasopharynx than French bulldogs and English bulldogs at level 1 and no differences were detected comparing the CSA of the tongue when normalized to total CSA (Table 4).

In some dogs, dorsal to the tongue, no airway except the endotracheal tube was detectable. This was observed at level 2 in three French bulldogs and at level 3 in one English bulldog, whereas this finding was noticed at level 2 in two and at level 3 in four of the pugs. Accordingly, some dogs had no detectable airway dorsal to the soft palate. At level 3, two French bulldogs and one English bulldog had no appreciable nasopharyngeal airway, while three pugs had direct contact of the soft palate with the dorsal nasopharyngeal mucosa.

### 3.3. Ratio of Total Air to Soft Tissue Areas

The ratio of total air to tongue area did not differ between the three breeds at all levels. Total air indexed to total soft tissue did not differ between the three brachycephalic breeds at levels 1 and 2 (Table 5), while at level 3, total air to total soft tissue was decreased in pugs compared to French bulldogs.

## 4. Discussion

The present study documents the relatively larger normalized total tongue volume in French and English bulldogs compared to pugs. The values of total tongue volume in French and English bulldogs presented in our study were very similar to previously reported dimensions for brachycephalic dogs (Table 2). Jones et al. demonstrated the presence of larger relative tongue volumes in brachycephalic breeds compared to mesaticephalic breeds [30]. However, normalized values of total tongue volume in pugs examined in the present study were significantly smaller than the median values in brachycephalic dogs and even smaller than those in mesaticephalic dogs, as reported by Jones et al. (Table 2) [30]. This discrepancy is possibly associated with the brachycephalic breed composition in the previous study, including eight French bulldogs, six English bulldogs, one pug, and one Boston terrier [30]. This preselection of breeds might have resulted in relatively high values of normalized total tongue volume representing the condition in French and English bulldogs than in brachycephalic dogs. Accordingly, the term macroglossia should be used cautiously in brachycephalic dogs, as in this study, pugs had smaller relative tongue volumes than other brachycephalic breeds. 

These findings add new knowledge to the breed differences in anatomical structures contributing to BOAS, such as the CSA of the nasopharynx, presence of nasopharyngeal turbinates, dimensions of the frontal sinuses, and incidence and degree of tracheal and bronchial collapse [10,26,32].

The second part of our hypothesis was partially rejected, as normalized CSA of the nasopharynx, soft palate, and tongue were only at some levels significantly smaller in pugs than in French and English bulldogs. Nonetheless, when normalized to skull length*skull width, nasopharyngeal and soft palatal CSA at level 1 were the smallest in pugs. These convergent relations led to equal relationships between airway and soft tissue CSA in these three brachycephalic breeds except for level 3 comparing pugs and French bulldogs. The comparatively larger CSA of the oropharynx at that level in French bulldogs is likely to result in a greater total air/total soft tissue ration than in pugs. 

From level 1 to 3, the total CSA followed the tapering anatomy of the naso- and oropharynx from rostral to caudal, with no significant differences comparing the three brachycephalic breeds at the same level. 

The CSA of the nasopharynx had been examined before, and our findings are partially in accordance with the previously published data [26]. While Heidenreich and colleagues detected significant differences in the nasopharyngeal CSA normalized to the LW index and bodyweight at the caudal end of the hard palate, but not at the caudal border of the tympanic bulla, significant differences were found in the current study at level 1 when normalized to total CSA and skull length × skull width [26]. It is likely that this discrepancy can be accounted for by the chosen reference (skull length × skull width and total CSA or bodyweight and LW index) and the location of the measurement, as in the present study, the level at the caudal end of the tympanic bulla was not assessed.

The two cranial selected positions of cross-sectional measurements were chosen based on previously published studies to provide comparable results. For instance, Heidenreich et al. measured the nasopharyngeal CSA directly at the end of the hard palate and caudal border of the tympanic bulla, whereas Liu et al. and Jones et al. recommended the caudal tip of the hamulus of the pterygoid bone [26,30,33]. The third level was chosen to assess the effect of macroglossia in these three breeds at a further caudally located position, as in a previous article, the smallest CSA of the nasopharynx was located at the caudal end of the soft palate [26]. The location cranial to the basihyoid bone was chosen to include the tongue in the measurements while maintaining a bone structure as a reference point. Interpretation of the results at this level must consider the possible effect of size differences between the breeds, representing a methodological limitation. However, the increasing slice thickness with increasing average weight of the breeds might partially account for slight relative differences in level positioning. Another limiting factor is the orientation of the hyoideus apparatus, which might be altered by the position of the head and the muscle tone of the jaws. Nonetheless, suspending the upper jaw in a neutral, open-mouth position is assumed to increase the likelihood of reliable and repeatable measurements [34]. 

The bodyweight was chosen to normalize the tongue volume to ensure reliability of data comparison, correcting a three-dimensional object solely to an accordingly defined reference. To account accurately for body size differences in the present study, the CSA were solely normalized to skull length × skull width and total CSA, enabling a two-dimensional normalization, that is less probably affected by body fat content than bodyweight.

The S-index (length of facial skull/length of cerebrum) has been proposed for comparing the skull dimensions, as it might correspond more accurately to the term brachycephalism than the LW index [9]. Accordingly, the reported LW index for pugs (median 0.88) and French bulldogs (median 0.89) were less different than the S-index (median 0.16 and 0.35, respectively), possibly affecting the results in the present study. However, the S-index was obtained on dorsoventral radiographs and adaption to multiplanar CT scans is still warranted. Moreover, comparison with existing studies would have been limited, if measurements would have been standardized using the S-index [9].

To the best of our knowledge, there are no studies describing experimental or clinically applied techniques of surgical correction for macroglossia in dogs affected by BOAS. Regarding potential recommendations for surgical correction of an overly large tongue, the findings of the present study outline the importance of accurately measuring tongue dimensions. In humans, macroglossia is related to a variety of congenital and acquired diseases, classified as related to tissue overgrowth, tissue infiltration, inflammation, and relative macroglossia [35]. A variety of entities such as Beckwith-Wiedemann syndrome, mucopolysaccharidosis, neurofibromatosis, and acromegaly may result in obstructive sleep apnea syndrome (OSAPS) [36,37]. In OSAPS, the pharyngeal airways are intermittently narrowed during sleep, and CT of patients affected by OSAPS reveal a significant reciprocal association between the apnea-hypopnea-index and the retrolingual cross-sectional airways [38,39]. Several surgical techniques have been proposed to address OSAPS. Among others, tongue suspension and reduction techniques have been introduced, such as hyoid advancement, genioglossus advancement, anterior wedge incisions, or posterior midline glossectomy [35,39,40]. Sleep-disordered breathing has been reported in brachycephalic dogs. Although extensive research is lacking and many dogs show comorbidities possibly contributing to sleep disorders, an obstructive component is suspected to be associated with high response rates to airway corrective surgery [23,25,41]. The contribution of macroglossia in brachycephalic dogs to sleep-disordered breathing and clinical signs of BOAS has not been evaluated and warrants further research. 

This study has several limitations. Owing to the retrospective nature of the study, positioning of the patients was not optimized for data acquisition, making small differences in the opening angle of the oral cavity and relative position of the tongue and soft palate. All dogs were intubated while acquisition of the CT scans, affecting the size of the oropharynx and might displace the soft palate dorsally, leading to a smaller CSA of the nasopharynx. Although this effect seems intuitive, a previously published prospective method control study revealed no significant difference in the nasopharyngeal CSA at the two levels examining the influence of endotracheal intubation in brachycephalic dogs [33]. Nonetheless, the caudal CSA of the nasopharynx in the transverse plane was 38% smaller in intubated dogs than in the non-intubated, and the CSA of the soft palate measured from cranial to caudal in the sagittal plane was significantly smaller in the intubated [33]. The definitive effect of intubation on soft tissue measurements was further elucidated, as the direction of variance of the soft palate thickness depended on the position of the endotracheal tube and the size of the caudal cross-sectional nasopharynx when extubated with larger areas being more affected by intubation than relatively smaller ones [33].

However, as all dogs in the present study were intubated, the effect would have evenly affected the measurements and repeated examinations in extubated individuals would less likely alter the main findings of this study significantly. The effect of endotracheal intubation on CSA and total tongue volume has not been examined in the veterinary literature; however, a significant impact on the latter appears unlikely.

Another limitation of this study is the static conditions displayed on CT scans, which cannot be accurately translated to the actual relation of soft tissues to the upper airways during the dynamic process of breathing cycles. Furthermore, the present study did not consider the grade of BOAS in affected animals, and no attempts were made to correlate the data to respiratory function tests [42].

The findings of the present study contribute to further refinement of the concept of macroglossia in brachycephalic breeds and its impact on the relationship of airways to soft tissues in the pharynx under non-dynamic conditions. Similar to other common reported findings characterizing BOAS, such as degree of stenosis of the nares, presence of aberrant nasal turbinates, nasal mucosal contact points, thickening and elongation of the soft palate, and everted laryngeal saccules, potential macroglossia should be subjectively assessed. The computational fluid dynamics may be utilized to define the impact of relative macroglossia on oropharyngeal and nasopharyngeal airflow, its contribution to increased airflow resistance in brachycephalic dogs and the effects of surgical interventions. Knowledge of a more precisely measured normalized tongue volume may be valuable in planning potential surgical correction methods in dogs affected by brachycephalic airway syndrome.

## 5. Conclusions

Tongue volume is smaller in pugs than in French bulldogs and English bulldogs, thus macroglossia is a condition encountered in selected brachycephalic breeds. Assessed values in pugs were comparable to previously published data in mesaticephalic breeds. Despite smaller tongue volume in pugs, comparison to French bulldogs and English bulldogs revealed similar total air to total soft tissue ratios. The data collected in this study can be used as reference for potential experimental cadaveric studies, investigating the effects of surgical procedures aiming to achieve tongue reduction in brachycephalic dogs.

## Figures and Tables

**Figure 1 animals-11-00662-f001:**
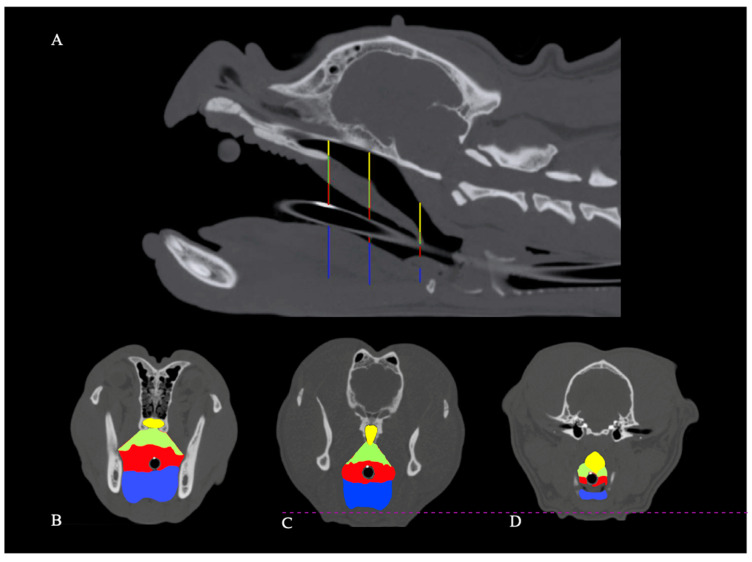
Computed tomography scans of an English bulldog skull. Vertical lines on the sagittal image of the skull (**A**) illustrate the position of corresponding transverse images of the same dog (**B**–**D**). B is located at the caudal tip of the hard palate, C is at the level of the last transverse slice with discernible humulae of the pterygoid bone, and D is positioned 5 slices cranial to the cranial tip of the basihyoid bone. Cross-sectional areas of the nasopharynx (yellow), soft palate (green), oropharynx (red; tracheal tube diameter not included), and the tongue (blue) are highlighted.

**Figure 2 animals-11-00662-f002:**
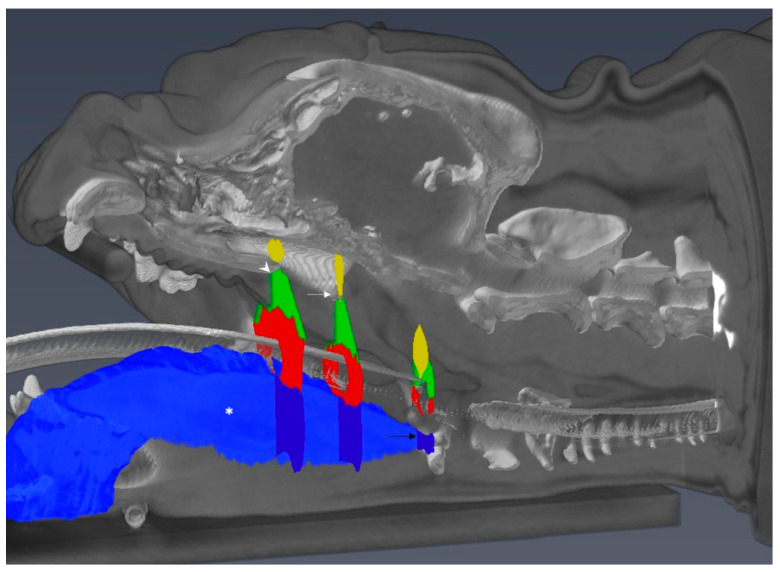
Reformatted 3D model of computed tomography scans of an English bulldog skull. Using segmentation software, the tongue volume has been assessed within defined borders (asterisk; depicted in shaded blue and sliced on a para-midsagittal plane for illustration purposes) and relative position of cross-sectional areas (white arrowhead: transverse plane 1 at the caudal tip of the hard palate; white arrow: transverse plane 2 at the caudal tip of the hamulus of the pterygoid bone; black arrow, transverse plane 3 cranial to the cranial border of the basihyoid bone).

**Figure 3 animals-11-00662-f003:**
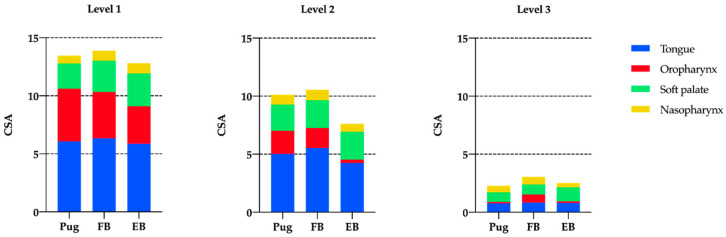
Median values of cross-sectional areas (CSA) of the nasopharynx, soft palate, oropharynx, and tongue normalized to skull length × skull width at level 1–3 in pugs, French bulldogs (FB) and English bulldogs (EB).

**Table 1 animals-11-00662-t001:** Comparison of age, bodyweight, and skull conformation between three brachycephalic breeds.

	Age (years)	Bodyweight (kg)	LW Index	Skull Length (mm)	Skull Width (mm)
Pugs ^m^ (range)	3.53 (1.9–12.44)	8.40 (6.0–11.0)	0.980 (0.89−1.06)	86.65 (75−100.8)	90.3 (79.9−102)
French Bulldogs ^m^ (range)	2.60 (0.65−8.95)	12.4 (7.8−16.6)	0.941 (0.83−1.05)	102.0 (84−111.8)	107.9 (94.9−119.4)
English Bulldogs ^m^ (range)	1.90 (0.69−7.59)	22.6 (14−33)	0.973 (0.89−1.02)	127.4 (106.4−142.7)	128.4 (105.7−144.4)

^m^ = Median

**Table 2 animals-11-00662-t002:** Comparison of tongue dimensions between three brachycephalic breeds.

	Pugs	French Bulldogs	English Bulldogs	*p*-Value			Brachycephalic Dogs *
				P vs. FB	P vs. EB	FB vs. EB	
Total tongue volume ^m^ (mm^3^) (range)	38,267 (30,450–53,981)	60,996 (46,362–88,871)	115,153 (60,432–140,396)	<0.0001	<0.0001	<0.05	68,352
Total tongue volume/bodyweight ^m^ (range)	4362 (3287–5268)	4936 (3830–7656)	4912 (4019–6348)	<0.0001	0.03	0.704	5650

^m^ = Median. * data published by Jones et al. 2019 [30].

**Table 3 animals-11-00662-t003:** Cross sectional areas of the nasopharynx, soft palate, oropharynx, and tongue normalized to skull length × skull width.

		Pugs ^m^	French Bulldogs ^m^	English Bulldogs
CSA nasopharynx/sl × sw	Level 1 Level 2 Level 3	0.678 ^FB,^ ^EB^ 0.879 0.603	0.879 ^P^ 0.906 0.668	0.903 ^P^ 0.700 0.403
CSA soft palate/sl × sw	Level 1 Level 2 Level 3	2.181 ^FB,^ ^EB^ 2.233 0.812 ^EB^	2.689 ^P^ 2.377 0.848 ^EB^	2.797 ^P^ 2.386 1.201 ^P,^ ^FB^
CSA oropharynx/sl × sw	Level 1 Level 2 Level 3	4.541 1.989 0.113 ^FB^	3.987 1.726 0.703 ^P,^ ^EB^	3.247 0.276 0.133 ^FB^
CSA tongue/sl × sw	Level 1 Level 2 Level 3	5.863 5.035 0.770	6.340 5.533 ^EB^ 0.832	6.064 4.261 ^FB^ 0.798

^m^ = Median, CSA = cross sectional area, sl × sw = skull length × skull width, Superscript letters indicate significant difference: FB = French bulldog, EB = English bulldog, P = Pug.

**Table 4 animals-11-00662-t004:** Cross sectional area of the nasopharynx, soft palate, oropharynx, and tongue normalized to total CSA.

		Pugs ^m^	French Bulldogs ^m^	English Bulldogs ^m^
CSA nasopharynx/total CSA	Level 1 Level 2 Level 3	0.053 ^FB,^ ^EB^ 0.084 0.199	0.064 ^P^ 0.085 0.205	0.066 ^P^ 0.088 0.131
CSA soft palate/total CSA	Level 1 Level 2 Level 3	0.170 ^EB^ 0.215 0.358	0.196 0.227 0.250 ^EB^	0.211 ^P^ 0.276 0.430 ^FB^
CSA oropharynx/total CSA	Level 1 Level 2 Level 3	0.338 0.177 0.048 ^FB^	0.291 0.144 0.224 ^P,^ ^EB^	0.264 0.039 0.06 ^FB^
CSA tongue/total CSA	Level 1 Level 2 Level 3	0.472 0.525 0.354	0.453 0.531 0.281	0.463 0.521 0.279

^m^ = Median CSA = cross sectional area Superscript letters indicate significant difference: FB = French bulldog, EB = English bulldog, P = Pug.

**Table 5 animals-11-00662-t005:** Summary of cross-sectional area ratios.

		Pugs	French Bulldogs	English Bulldogs	*p*-Value		
					P vs. FB	P vs. EB	FB vs. EB
Total air/tongue ^m^	Level 1 Level 2 Level 3	0.813 0.497 0.894	0.737 0.382 1.866	0.700 0.270 0.639	0.585 0.310 0.090	0.633 0.310 0.090	0.994 0.310 0.090
Total air/total soft tissue ^m^	Level 1 Level 2 Level 3	0.611 0.363 0.641 ^FB^	0.536 0.272 0.910 ^P^	0.480 0.146 0.240	0.409 0.181 0.045	0.424 0.181 0.999	0.976 0.181 0.115

^m^ = Median CSA = cross sectional area Superscript letters indicate significant difference: FB = French bulldog, EB = English bulldog, P = Pug.

## Data Availability

The data presented in this study are available in supplementary material.

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
