# Peer review of "Tongue and Upper Airway Dimensions: A Comparative Study between Three Popular Brachycephalic Breeds"

_animals, 2021, doi:10.3390/ani11030662_

Round 1

Reviewer 1 Report

Tongue and upper airway dimensions: response to revision

I would like to thank the authors for revising the manuscript. Many parts are more understandable now. However, I must again criticize the way of normalization of tongue volume and CSA.

The authors believe, that the LW index would be appropriate to normalize morphometric data form different sized dogs. The LW index as other indices is meant to describe a relation of several distances, here the degree of brachycephalism. Interesting enough, the LW index did not really differ between P, EB and FB in your study (see table 1), which is rather normal. All breeds are brachycephalic. The LW index does not refer to the mass or size of the individual animal and can therefore not be used for normalization. The same is true, if you normalize volumes  in different sized dogs by length.

Let me give you an example: a child (100 cm height, 12.5 kg, tongue 5 x 5 x 5 cm ) and his father (200  cm height, 100 kg, tongue 10 x 10 x 10 cm) should have the same normalized tongue dimension. Volume calculation of the tongue is:  child 125 cm3, father 1000 cm3. Now tongue volume normalized by height is: child 1.25 cm3/cm, father 5 cm3/cm.  You see: the normalization using a single length dimension is wrong. If you normalized by BW, the tongue volume amounts 10 cm3/kg for both  as expected. You see: the method of normalization has major impact to the outcome. All are mathematically correct, however the test method is not appropriate if you do not normalize volume by volume (or at least BW, which somehow reflects volume).

The normalization method proposed by Jones et al. (Vet Surg 2019), using indices or lengths to normalize volumes or CSA is wrong. The fact, that the paper was published, does not necessarily mean, that the methodology must be correct. Please do not do the same error. I herewith clearly disagree to your response to my comment regarding normalization, even if a mathematician has given advice. Be aware, that I also have chosen support for my comments herein by consulting a bioengineer, who has tremendous experience in comparing data between different sized animals.

I again recommend major revision regarding the normalization procedure and I suggest to normalize tongue volume only to BW and not to indices or length. You have found some interesting observatioins with normalization to BW, which can be used throughout the paper. Furthermore, please compare CSA only to length* length data (e.g. skull length*skull length, or length*width) or to BW2/3. All other normalization calculations (e.g. CSA to length only) must be deleted from your manuscript. This makes the manuscript much more readable. And the results are different than those you discussed.

In addition to that, a more simple and visual presentation of your data in table 4 in a column would be of great value, also because the proportion covered by the airways becomes immediately visible. The reader would then easily see, that e.g FB have larger orophyaryngeal areas at level 3 than P or EB.  However, the data of Table 4 needs some verification, especially to explain/complete the shortage in total CSA (sum of all reported partial CSAs) at levels 2 and 3.

The whole discussion needs some rearrangements, when the data are recalculated. If the review process allows it, I would send you some clarifying graphs by pdf in an attachment.

In summary, I can not support publishing the manuscript with  distinct methodical errors leading to improper conclusions. In case, the authors again disagree with my criticism about normalization, I recommend to the Editor in Chief to advise a different reviewer.

Author Response

Thank you for your time and the effort, providing substantial contribution in order to improve the methodology of this article.

Just for clarification, we understand basic math, being totally aware, that mixing up dimensions when normalizing will affect data significantly. However, so far no study has determined the "best" way of normalization and initially we were concerned, that using an approach of normalizing cross sectional areas in the skull, that had not been reported so far (in small animal medicine at least), would hamper comparability to recently published articles. In addition, correction of cross sectional areas by bodyweight (even to the power of 0.67) will have the same flaws as skull length due to the dimension and interindividual differences in body fat content and body configuration. Moreover, the product of skull length * skull width, does not necessarily reflect the relevant configuration of the area of interest, as brachycephalism affects the facial skull differently than the neurocranium (which has been shown by Koch et al., establishing the S-index). Unfortunately, using CT-data, the S-index, which has been defined for plain radiographs cannot be recommended yet for normalization. None of these circumstances makes it any less true, that normalization of a square by a line will shift datasets and the article might be less confusing in the current state, as dimensions of normalization are logical and homogeneous.

Accordingly, the authors have deleted the sections including other normalization methods than bodyweight and two-dimensional terms. As the authors consider data, presented in tables a relevant reference for future studies by other working groups, we decided to keep the tables. Another figure has been added, hence, we hope that the key message of this section can be easily apprehended, whereas no discrete information is lost.

The relevant passages of the discussion have been changed accordingly.

Reviewer 2 Report

The paper is improved by the revisions- thanks you

I still think the tongue volume measurement is pretty robust and the oropharyngeal cross sectional area much less so (different opening angles, intubated, resting of the head on a bean bag that may allow the pharyngeal tissues to compress). Although this is addressed in the limitations many readers do not get this far....

Author Response

Thank you for your time and comments, helping to improve this research project.

This manuscript is a resubmission of an earlier submission. The following is a list of the peer review reports and author responses from that submission.

Round 1

Reviewer 1 Report

General remark

This manuscript shows a novel and interesting approach to the role of the tongue and possible macroglossia in the brachycephalic obstructive airway syndrome. The general outline of the study is well chosen and would contribute to the gain of knowledge in this disease.

My major concern in this study is the method of normalization. Although used in other studies, a normalization to compare different sized animal by dividing  a three-dimensional item (like tongue volume) by a one dimensional item (like skull length) carries the high risk of methodological failures. If ever, volume is normalized to BW (also a three-dimensional parameter) and, cross sectional area to another section (but not length or index). As you compare different sized  breeds, the error in normalisation could be the origin of your difference. As I am not a physicist, I would recommend to consult a specialist to clarify such questions and recheck the data afterwards. My conclusion from the data available is, that the tongue volume is not different between the 3 breeds, because the values normalized to BW did not differ. And if this is correct, the whole discussion must be rewritten.

Another major question comes from the positioning of the dogs. As mentioned in the manuscript, the study is of retrospective manner. Therefore, mouth opening is surely not standardized and the palate not parallel to the ground. This can be seen on Fig. 1 with the palate oblique to the ground. CSA therefore were not obtained perpendicular to the palate and therefore carry the risk of being to large. Furthermore, mouth openings and different sized oronasal tubes must influence the CSA of the free air in the naso- and oropharynx. I beg the authors to clarify, how and if they standardized these measurements.

There are many findings not discussed. The authors could have compared the normalized values to the LW or S Index. This would have given a idea, how the skull shape influences the CSA of the tongue, the palate and the free air. I admit, that was not the goal of the study, but It could contribute to a better understanding of the BOAS

Specific comments

57        missing brackets on reference 18

83        Please clarify how you came to the hypothesis, that French and English Bulldogs would have larger tongue volume than pugs. The reader cannot understand, why you assume this before you conducted the study. It is not the idea, that the result of the study is the hypothesis !

91        BOAS was already introduced, so please use it instead of the whole expression

110      The palatum should be parallel to the table – however Fig. 1 shows something different

130      missing subfigure C in Fig. 1

135      Please explain, why the incisor teeth (and not the tongue margin) was choosen as a delineator

149      5 CT slides is not a normalized distance when you compare different sized animals.

161      According to the text, the reader expects some more anatomical landmark in the Fig. 2, please add some arrows and text to Fig 2

170      as mentioned above, the normalization method is not clarified and not discussed in the discussion part

188      I would recommend to move the data of the dogs to the material&methods section and add a subchapter “dogs”

198      Table 1 is not of major interest and the P-values of age, BW and others not interesting for the study purpose nor discussed later; please shorten

204      If the data normalized to BW are correct, the whole discussion must be rewritten. I lack a critical word on the different methods of normalization and the conclusion out of it. The reader does not know, in what data he should trust, therefore:

245      The first sentence of the discussion section is doubtful and the reader does not know, if the tongue volumes differ or not

259 ff  Interesting first sentence. However, I would like to read something about the role of tongue volume to the pathophysiology of BOAS, whether the tongue presses onto the soft palate or it is the result of negative pressure or result of edema. There is plenty of room to discuss the problem for the clinician.

429      Double (23)

Reviewer 2 Report

Interesting paper.

I think the title and abstract need to clarify that these results have been found in BOAS affected dogs that are intubated and in an open mouthed position.

Although this may sound pedantic I think it is likely that different results may have been found (particularly in the oropharyngeal dimensions, which I would remove) if the mouth was closed and the dogs were extubated.

In the methods the open angle of each CT does need measuring and documenting and the size of ET tube should be recorded.Alternatively remove the air measurements (oropharynx) as I think the tongue measurements are likely to be consistent and the palate potential variation in measurement is addressed in limitations (though that paper referenced and used as a comparison had a fixed spacer used).

Specifically

Line 66-67 slightly strange to discuss clinical findings in Frenchie and then flip to anatomical changes in pugs. Lack of frontal sinuses is not (yet) linked to BOAS and found in other breeds.

131 - previously reported- need to clarify this was an extubated paper

109- expand methods- mouth opened to what degree? spacer used? or measure open angle and record

Table 3- what happened to CSA oropharynx level 2 bulldog. Text and table do not seem to correspond?

Can we have similar image of pug and French bulldog

Overall good project but the results are over ambitious considering the limitations of the retrospective intubated study. I would stick to measurements that can be assessed confidently given these limitations as I think this would make for a much stronger paper where the macroglossia main message is still valid.

Reviewer 3 Report

I quite enjoyed this paper, and have appreciated the contributions this team has made to the scrounge of brachycephalic airway disease. I have no significant suggestions or areas for improvement. 

Author Response

Thank you for your time revising the research manuscript.

Reviewer 4 Report

These authors studied tongue volume to assess the presence of macroglossia which can cause the obstructive airway syndrome seen in brachycephalic dog breeds. Three breeds were compared using cross-sectional computerized tomography screens at 3 tongue locations of Pugs, French Bulldogs and English Bulldogs. Perhaps surprisingly, the tongue was smaller in Pugs than in the other 2 breeds, and at its most caudal part French bulldogs had a larger oropharyngeal area than the other 2.  The authors concluded that the term macroglossia should not be used to describe the Pug breed.  Surgical correction should consider this finding.

Author Response

(The authors gave the same response as above.)
